# Direct imaging of electron density with a scanning transmission electron microscope

Ondrej Dyck [1,4] ✉, Jawaher Almutlaq [2,4], David Lingerfelt [1,4], Jacob L. Swett[3], Mark P. Oxley [1], Bevin Huang [2], Andrew R. Lupini [1], Dirk Englund [2] & Stephen Jesse[1]

Recent studies of secondary electron (SE) emission in scanning transmission electron microscopes suggest that material's properties such as electrical conductivity, connectivity, and work function can be probed with atomic scale resolution using a technique known as secondary electron e-beam-induced current (SEEBIC). Here, we apply the SEEBIC imaging technique to a stacked 2D heterostructure device to reveal the spatially resolved electron density of an encapsulated $WSe_2$ layer. We find that the double Se lattice site shows higher emission than the W site, which is at odds with first-principles modelling of valence ionization of an isolated $WSe_2$ cluster. These results illustrate that atomic level SEEBIC contrast within a single material is possible and that an enhanced understanding of atomic scale SE emission is required to account for the observed contrast. In turn, this suggests that, in the future, subtle information about interlayer bonding and the effect on electron orbitals could be directly revealed with this technique.

Secondary electron e-beam induced current (SEEBIC) imaging using a scanning transmission electron microscope (STEM) can reveal material properties such as electrical conductivity, connectivity, and work function[1,2]. STEM-SEEBIC imaging relies on the emission of secondary electrons (SEs) from a sample induced by primary beam electrons. The accumulation of positive charge on the sample, from repeated emission of negatively charged electrons, is dissipated by grounding the sample through a transimpedance amplifier (TIA) which measures the electron current flowing into the sample. Thus, electronic properties of the sample can be explored with the resolution offered by modern STEMs. SEEBIC imaging has been used to reveal the filamentation and dielectric breakdown involved in the switching processes in valence change memory devices[3]. It has also been used to detect conductance switching in graphene nano-gaps[4] and to enable resistive contrast imaging in STEM where image intensity is directly related to sample conductivity[1,5,6]. Lattice resolution has been demonstrated in a 3D crystal to the angstrom scale[7] as well as the detection of different layer numbers of graphene[8].

SE emission is typically described from the perspective of a macroscopic material, leveraging ensemble material properties such as dielectric function. However, these descriptions fail to describe emission (ionization) at the atomic scale due to the statistical nature of the approach. In this work, we show experimental atomically resolved SEEBIC images that exhibit contrast differences between adjacent lattice sites in the same material. To understand both the atomic resolution and variation in contrast one must abandon the macroscopic material description of SE emission, turning to first principles atomistic ionization modeling. With this framework we conclude that our image contrast is directly proportional to the projected sum of electron orbital ionization cross sections. The SEEBIC image, then, represents a projected view of the sum of the electron orbitals weighted by the ionization probability of the electrons in each orbital. Stated differently, the SEEBIC image reveals the electron density distribution in the material based on the atomic ionization cross section, viewed in projection. Moreover, we illustrate that a model designed to describe beam-induced ionization of valence electrons in a simplified

[1]Center for Nanophase Materials Sciences, Oak Ridge National Laboratory, Oak Ridge, TN, USA. [2]Massachusetts Institute of Technology, Cambridge, MA, USA. [3]Biodesign Institute, Arizona State University, Tempe 87287 AZ, USA. [4]These authors contributed equally: Ondrej Dyck, Jawaher Almutlaq, David Lingerfelt. ✉e-mail: dyckoe@ornl.gov

WSe₂ structure is insufficient for fully replicating the contrast observed from encapsulated WSe₂ leading to the conclusion that this imaging mode could possibly be used to capture subtle changes in the electron density distributions from the effects of, in this case, interlayer interactions.

Previous efforts to reveal electron orbital information using (S) TEM-based techniques have not gone unrewarded. A variety of electron energy loss spectroscopy (EELS) experiments have successfully been used to reconstruct electron orbitals or tease out bonding information from carefully crafted experiments[9–11]. Likewise, convergent beam electron diffraction (CBED) techniques have similarly been used to extract electron orbital information[12]. Beyond the STEM, in scanning tunneling microscopy (STM), the primary imaging mode directly probes electron orbitals on the sample surface routinely producing stunning images of the electronic structure of materials at high resolution (see for example refs. [13–15].). Experiments using femtosecond laser pulses have shown tomographic imaging of electron orbitals with attosecond tracking of the electron wave packet dynamics[16]. X-ray photoelectron spectroscopy can also be used to directly image atomic valance orbitals[17].

With all of these impressive demonstrations one might get the impression that the imaging of atomic orbitals is now a matter of routine. With the exception of STM imaging, this is not the case. What makes STM imaging unique in this regard is that the primary contrast generation mechanism is the electronic structure of the specimen surface. In STEM-EELS, for example, the signal is generated by energy lost from a primary electron which can arise from any possible energy transition within the material. The EELS spectrum, therefore, represents a mixture of different signals that are often challenging to separate and interpret. Likewise the femtosecond laser tomographic imaging and X-ray photoelectron spectroscopy are likely just as challenging as STEM-EELS if not more so, due to difficulties unique to their respective experimental modalities.

Since STEM-SEEBIC exclusively reports on ionization events, it can provide information not usually accessible via standard EELS. The onset of ionization for the most weakly bound valence electrons in materials can overlap energetically with other non-ionizing excitations such as interband transitions and plasmonic losses, preventing discrimination of primary electron energy losses due to ionization events. Furthermore, since the primary electron energy losses report only on the initial excitations enacted through inelastic scattering, information on secondary electrons generated through internal electronic reorganization processes occurring after the initial excitation (e.g., Auger processes) is not present in the EELS spectrum. Thus, while the total secondary electron yield is provided directly via the SEEBIC intensity, the totality of information needed to produce this same quantity is not accessible with knowledge of the primary electron energy loss function alone, regardless of how it is measured/analyzed.

Here, we show that the STEM-SEEBIC contrast generation mechanism, as in STM, can be interpreted as a direct measure of the electron density of the specimen. The SEEBIC imaging modality is insensitive to nuclear scattering. Likewise, non-ionizing electronic transitions (e.g. plasmons, interband transitions) are also not detected. The only source of contrast is the successful emission of electrons arising from ionization. Because the STEM affords high resolution, contrast variation in this mode can be attributed to variation in ionization cross section with e-beam position (convolved with the point spread function of the imaging system). This means that atomically resolved SEEBIC images represent a sum of the electron orbital ionization cross sections, viewed in projection. Thus, the electron density is not uniformly represented in the SEEBIC image, but is modulated by the terms in the ionization cross sections associated with binding energy of the electrons relative to the Fermi level and the coupling strength between the initial and final electronic states due to the electric potential associated with the incident high-energy electrons.

The core-shell electrons in this case have binding energies of the order of keV and very small ionization cross sections compared to the outer/valence electrons which have binding energies of 10–100 s of eV.

## Results and discussion
### Sample fabrication and geometry
To acquire STEM-SEEBIC images an electrically conductive pathway to the device must be made and the device must reside on an electron transparent substrate (ideally suspended). To satisfy these constraints, custom devices were fabricated featuring lithographically patterned electrodes aligned with an electron transparent window after which holes were milled using a focused ion beam (FIB). A detailed report of the device fabrication process can be found in a prior publication[4]. Briefly, a 300-μm-thick Si base with a 1000-nm-thick thermal oxide layer and 20 nm of low-stress LPCVD SiN$_x$ formed the base substrate. Cr/Au metal electrodes were lithographically patterned and deposited on the surface to facilitate electrical connections from the STEM holder to the 2D heterostructure device. Back-side etching was performed using KOH at 80 °C to form electron transparent windows beneath the ends of the metal electrodes. Apertures were FIB milled between the electrodes to provide a region over which the 2D heterostructure would be fully suspended.

The heterostructure device was assembled following a dry transfer method, described in the "Methods" section. Cross-section and plan-view diagrams of the device structure on the SiN$_x$ window are shown in Fig. 1a, b, respectively. An optical image of an as-fabricated device is shown in Fig. 1c with the exfoliated 2D flakes highlighted by the colored, dashed lines. The layer order is shown in the inset.

STEM imaging was performed using a Nion UltraSTEM 200 and is described more fully in the "Methods" section.

### Overview comparison of HAADF and SEEBIC imaging
Figure 2 shows overview HAADF and SEEBIC images of the suspended 2D device acquired at 100 kV. In the HAADF image, Fig. 2a, we can see the end of a metal electrode and the FIB milled aperture, over which the device is suspended. Figure 2b shows the corresponding SEEBIC image which was acquired simultaneously with the HAADF image. Bright regions are electrically conductive and electrically connected to the TIA. This allows a clear delineation of the edge of the graphene layer (marked in the figure) which is not visible in the HAADF imaging mode. The rest of the 2D layer stack (h-BN/WSe₂/h-BN) appears dark, indicating it is non-conductive and/or electrically isolated from the TIA. In another region of the aperture, the edge of the WSe₂ layer can be observed based on the Z-contrast of the HAADF signal[18], shown in Fig. 2c. In the top half of the image the WSe₂ layer is missing so we have two h-BN layers followed by a graphene layer on the surface. In the bottom half of the image, we have our full device stack (graphene/h-BN/WSe₂/h-BN). The inset shows an atomically resolved image of the WSe₂ edge. Additionally, the h-BN can be detected using core-loss electron energy loss spectroscopic (EELS) imaging (an example is shown in the Supplementary Information Note 2). Thus, this suite of characterization modes enables identification of each material in the stack.

### Atomically resolved imaging
In Fig. 3, we examine an atomically resolved, sequentially acquired HAADF image stack with a simultaneously acquired SEEBIC image stack. Five image frames (128 × 128 pixels) were acquired in succession with a pixel dwell time of 8.2 ms (134 s/frame) at an accelerating voltage of 80 kV and a nominal beam current of 160 pA ($1.3 \times 10^{11}$ electrons/frame). Single frames from the HAADF and SEEBIC channels are shown in Fig. 3a, b, respectively. The images were acquired on the full heterostructure stack, graphene/h-BN/WSe₂/h-BN, however, only the WSe₂ signal is evident in the HAADF signal. This is due to the much stronger electron scattering of the heavier W and Se atoms. The SEEBIC

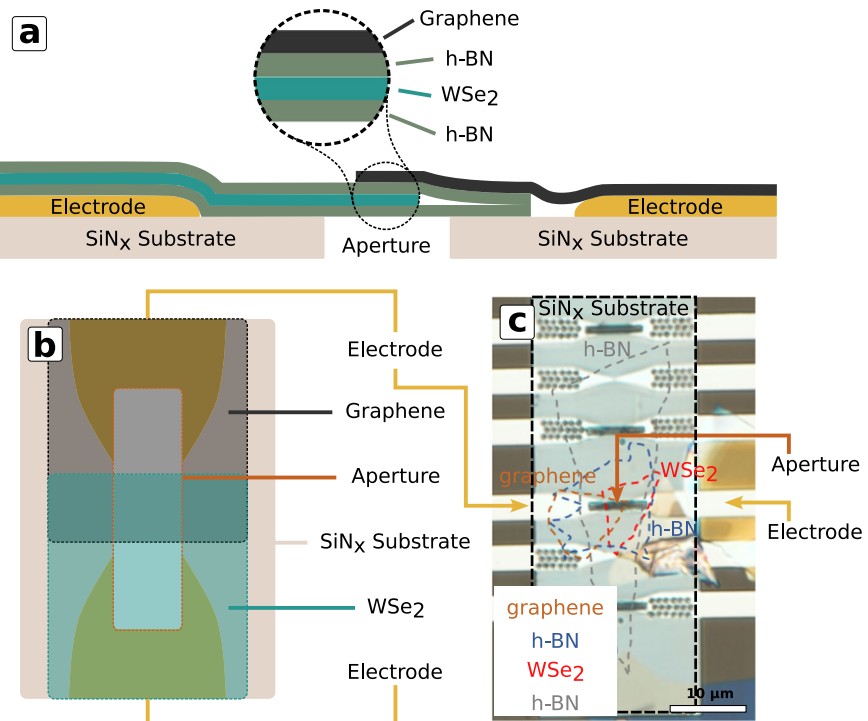

**Fig. 1 | Schematic and optical image of 2D heterojunction stack. a** Cross-section schematic view of suspended device structure. **b** Schematic plan view of device structure. **c** Optical image of the as-fabricated device. Approximate positions of the layer locations are outlined in color-coded dashed lines. Layer order is shown in the inset (substrate is on the bottom).

response is much more uniform and noisy, nevertheless, it appears that it also contains atomically resolved information.

There are multiple layers in the heterostructure stack so we expect that the observed response is a mixture of signals originating in each layer (see Supplementary Information Note 1). These layers were not deliberately aligned with each other or with the WSe$_2$, thus creating several periodic background signals. The features contained within the SEEBIC image are therefore difficult to directly analyze. Instead we use an analysis approach that leverages the Z-contrast information contained in the HAADF image. A deep convolutional neural network (DCNN), trained to find carbon atoms in graphene[19], was used to estimate the pixel-level probability of being an atom for each HAADF image in the five-frame stack. The output probability map, labeled 'DCNN' in Fig. 3c, was thresholded and the center of mass of each blob was taken as the atom locations, labeled 'Atom Finding'. Image tiles were cropped from both the HAADF and SEEBIC image stacks centered on the atom locations. A few example HAADF tiles are shown, labeled 'Tiling'.

To discriminate between the W and Se−Se lattice sites, k-means clustering was performed on the HAADF image tiles using two clusters. This approach uses information contained in close proximity to the lattice site, including neighbor atom intensities and rotational symmetry, for a more robust decision-making feature set than that afforded by intensity alone. The k-means cluster labels for one frame are plotted as a color-coded overlay on the HAADF image, labeled 'k-means Clustering'. Based on the clustering labels, histograms of individually observed atomic intensities (mean value over a 4-pixel radius from the atom position) were assembled for the HAADF stack, Fig. 3d, and SEEBIC stack, Fig. 3e. Here, we can clearly see the intensity overlap in the HAADF signal between the W and Se−Se lattice sites, which are not separable using intensity alone.

Finally, the mean response of W and Se−Se lattice sites for HAADF and SEEBIC is shown in Fig. 3f, g, respectively. These images were generated by taking the mean of all the tiles in each category. Because the tile selection has been aligned to the atomic positions of the WSe$_2$, any other features generated by the rest of the heterostructure stack (the h-BN and graphene) are smeared out into an average background. The intensity in the images shown in Fig. 3g has been shifted such that the minimum value in the mean image, which is our best estimate of the average background signal, is equal to zero. This allows the image intensity to be interpreted directly as a current relative to the background. The minimum and maximum of the displayed intensity were determined by selecting the minimum and maximum values found in the mean images. In this way, the relative intensity of the images is preserved and the minimum value is set to zero. The colorbar lists the mean measured current values in pA. Likewise, the current values shown in Fig. 3e have been rescaled to match. The mean values are marked with vertical lines, the width of which correspond to the mean standard error. This is worth further comment. The intensity variation observed in the SEEBIC histograms is relatively large. However, this analysis does not rely on the SEEBIC data for the discrimination of the atomic positions or intensities. Instead we rely on the much more clearly resolved HAADF image. Once this information is available, we can produce reliably selected tiles from the SEEBIC data and preserve our ability to discriminate between the lattice site types based on the prior labels. When performing the calculation of the mean response, our uncertainty is no longer represented by the standard deviation of the original data distributions (shown in the histograms) but the error in the mean (width of the vertical lines) which is greatly reduced by the number of samples, producing a much higher confidence for the mean as compared with the individual data points. Based on this analysis, we find that the mean SEEBIC emission rate for a Se−Se lattice site was greater than for a W lattice site. A Python Jupyter notebook that reproduces these results can be found at https://github.com/ondrejdyck/SEEBIC_electron_orbitals.

It is worth pointing out that, despite the involved approach to analyze the data shown in Fig. 3, all we are doing is producing an average. Complications have arisen exclusively because it is difficult to

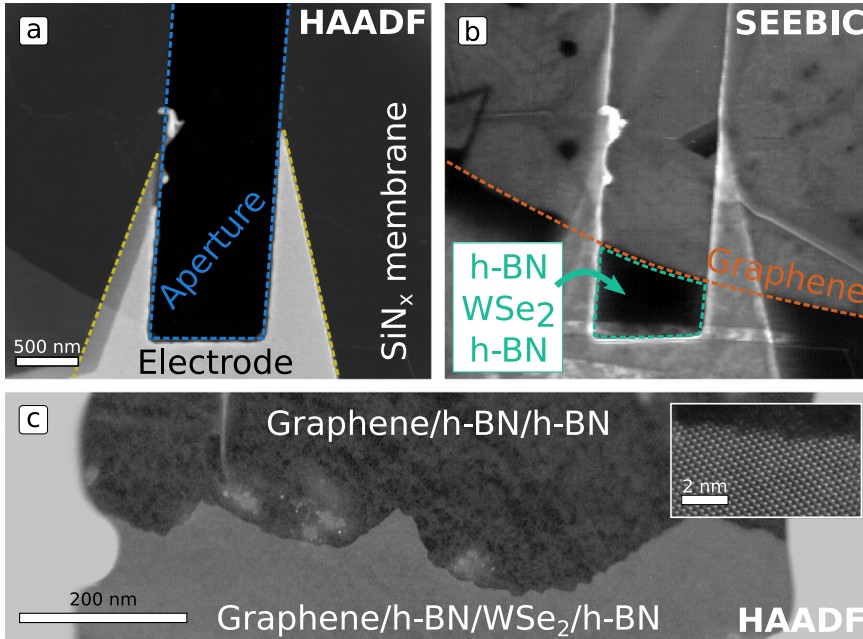

**Fig. 2 | Overview high angle annular dark field (HAADF) and secondary electron e-beam induced current (SEEBIC) images of the 2D heterostructure device.** **a** Overview HAADF image with various major features labeled. The aperture and electrode are marked by blue and yellow dashed lines, respectively. **b** Overview SEEBIC image acquired concurrently with the HAADF image shown in (**a**). Nominal beam current was 68 pA. The electrically conductive graphene layer can clearly be observed as brighter than the other regions, marked by the orange dashed line. The suspended h-BN/WSe$_2$/h-BN stack is marked by the green dashed line. The scale bar in (**a**) also applies to (**b**). **c** Overview HAADF image of the edge of the WSe$_2$ layer. The WSe$_2$ layer appears brighter due to the Z-contrast in this mode. A higher resolution image of the WSe$_2$ edge is shown inset.

tell the computer where the atoms are and how to distinguish W from Se–Se lattice sites, so that we do not mix our signals. With these details addressed, the images shown in Fig. 3f, g were obtained by a simple average of the raw tiled data. The zero point used for the color bar in the SEEBIC images has been set manually to the minimum value found in the images, however, this does not change the relative intensity variations observed.

The HAADF images shown in Fig. 3f represent the nuclear scattering cross section of the material, convolved with the point spread function of the instrument. HAADF imaging is attractive in this regard because it is easier to interpret[18] than bright field imaging, which contains much more information. It is precisely because the bright field image contains more information that it becomes difficult to interpret.

The SEEBIC images shown in Fig. 3g represent the total ionization cross section of the material, convolved with the point spread function of the instrument. The physics of SEEBIC image acquisition dictate that nuclear scattering or electronic transitions that do not result in ionization are not observed. Thus, the SEEBIC image represents the total electron density of the specimen contributed by each of the occupied electronic states (modulus squared) summed together, weighted by their respective ionization probabilities, and viewed as an image projection. Much of the information on beam-induced ionization is also present in EELS data, which detects energy losses of the primary beam electrons, however, many other energetic transitions are simultaneously detected with EELS. This makes the interpretation of secondary electron yields from the EELS signal more difficult. In addition, the description of ionizing excitations in EELS is diminished by high angle scattering which is not captured, and the inherent inability to quantify secondary electron yields from Auger processes. In contrast, the SEEBIC technique does not rely on an external electron detector and so has a detection efficiency of effectively 100%[2].

In light of these comments, it is worth explicitly pointing out that reabsorption of electrons can also occur. An SE emitted by an atom can be recaptured by the surrounding material resulting in zero net emission and this is therefore not detected in the SEEBIC signal. This phenomenon is typically captured theoretically by using a parameter called the 'escape depth', which decays exponentially with the emission depth in the material[20,21]. The physical justification for the use of such a parameter is obvious, however, the derivation of such a parameter from a bulk, homogeneous, and continuous material raises questions about the applicability to highly localized and atomically resolved images of 2D materials, where the information being observed is discontinuous and heterogeneous in nature. Nevertheless, the assumption that this reabsorption process has some influence over the observed image intensity is well grounded, the details of which should be explored in the future. Here, we merely point out that any atomic scale reabsorption variations arising from, for example, emission direction and the crystallographic orientation of the overlayers, is mitigated in large part by the averaging procedure due to the larger area represented by the sampling.

## Theoretical approach to atomically resolved SEEBIC

This brings us to a more general concern with the understanding of SEEBIC image intensity. We have presented the SEEBIC intensity, thus far, from the view of individual ionization events. But, historically, this has not been the standard approach to treating this phenomenon. The theory of secondary electron (SE) emission from a variety of materials and primary beam energies has been studied from a macro-scale (that is, non-atomistic) perspective for understanding, among other things, the contrast observed in scanning electron microscopes (SEMs)[20,21]. These treatments leverage material properties such as work function, Fermi energy, mean free path length, etc. that consider a material as a single uniform block and were developed, generally speaking, to describe thick materials where the primary electrons (or other high-energy particles) lose and transfer energy as they propagate through the material. Clearly, such descriptions begin to breakdown in the limit of 2D materials and with atomically resolved experiments where the

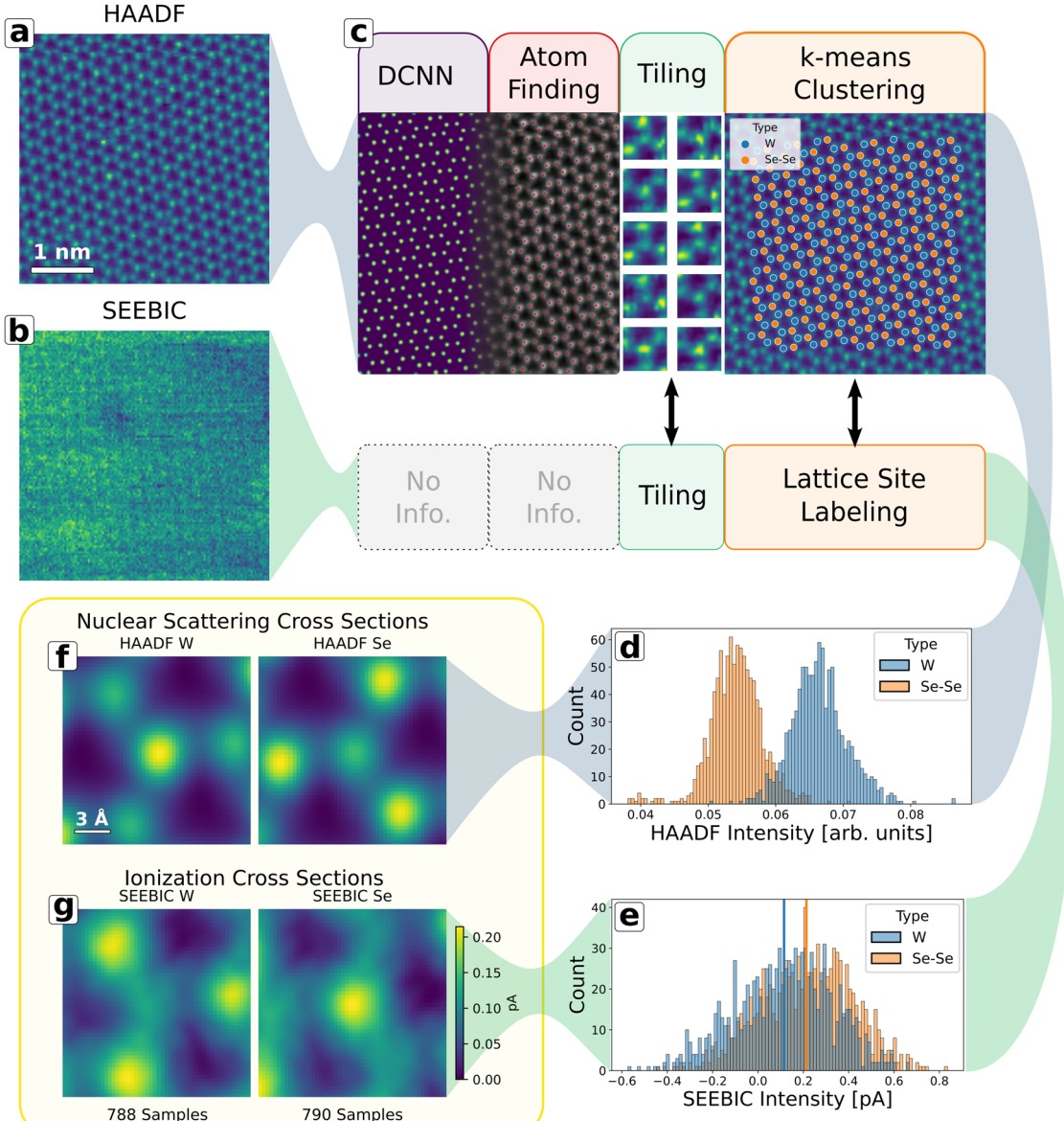

**Fig. 3 | Summary of image processing and intensity analysis workflow.** Single frames from a simultaneously acquired HAADF (**a**) and SEEBIC (**b**) image stack acquired through the full heterostructure device. The scale bar from (**a**) also applies to (**b**). **c** Depiction of image processing workflow: a deep convolutional neural network (DCNN) was used to identify atomic positions in the HAADF image, image tiles were extracted centered on the atomic positions for both the SEEBIC and HAADF images, k-means clustering was used to identify the lattice sites using the HAADF image tiles. Histograms of the lattice site intensities for HAADF and SEEBIC are shown in (**d**) and (**e**), respectively. Histograms are color-coded according to lattice site type. The mean atomic response for the HAADF and SEEBIC signals is shown in (**f**) and (**g**), respectively, separated according to lattice site type. The scale bar in (**f**) applies to all four panels in (**f**) and (**g**). The SEEBIC current for (**g**) and (**e**) is scaled such that the minimum response from both tiles in (**g**) was set to zero. All colored images were colored using the viridis color map in matplotlib.

emission from a single atom can be directly compared to the emission from its neighbor in the same material. This is the same issue that arises when considering escape depth at an atomic length scale.

To properly account for the measured image intensities, our theoretical treatment must abandon bulk descriptions and begin to treat the materials atomistically. The SE yield originating from inner-shell electrons in the material has been treated successfully using the corresponding ionization cross sections for the isolated atoms of the same type[22]. While these electrons contribute to the contrast in the SEEBIC images, they cannot be invoked to capture the SEEBIC intensity contributed by the ionization of bonding electrons, which are being preferentially selected by irradiating in the interatomic regions. Here, we begin this process and describe an approach that has the potential to properly account for the observed intensities at the sub-atomic

length scale. To be clear, we will not attempt to show a direct match with the experimental data here, nor will we attempt to draw any firm conclusion. The objective is to illustrate the direction one must take to properly account for the physical processes underlying the experiment, for which modeling the SEEBIC intensity involving only the inner-shell ionizations, as well as the classical, bulk treatment fails. This reconceptualization of the generation of SEEBIC image contrast is what has led to the present interpretation.

As a first approximation to the ionization rates due to irradiation at a given lattice site, one may consider total ionization cross sections calculated for the individual atoms that comprise the material. Approaches such as the binary encounter dipole approximation[23,24] have historically been used to this end, and produce atomic ionization cross sections in good agreement with those measured

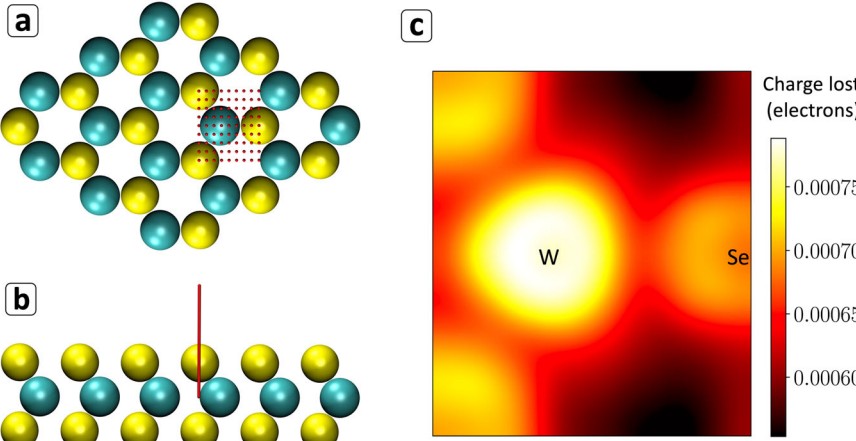

**Fig. 4 | Summary of simulated ionization.** Valence ionization rates resolved as a function of position of a beam-like perturbation from quantum electronic dynamics simulations with absorbing boundary conditions. For each of the positions indicated by red dots in the plane of the material (**a**) separate time-dependent density functional theory (TD-DFT) electronic dynamics simulations were carried out after subjecting the system to an electric field impulse corresponding to the average electric potential associated with a line of 100 evenly-spaced electronic point charges extending 10 Bohr radii from the indicated positions (in the W atomic plane), as illustrated by the red line in (**b**). Green spheres represent W; yellow spheres represent Se. The total charge lost to the absorbing boundary conditions upon perturbing at each position indicated in (**a**) is plotted as a (Lanczos-interpolated) heatmap in (**c**).

experimentally. While the ionization probabilities for very deep core levels in high Z elements will be essentially identical for the isolated atoms and those same atoms residing in a material, hybridization of the valence orbitals can be expected to nontrivially affect the ionization rates for any electrons in materials that are engaged in bonding. In comparison, the HAADF signal is formed mainly by beam electrons scattered to high angles by atomic nuclei and the low-loss (valence) EELS signal reports on the excitation of delocalized bosonic quasiparticles and collective electronic modes; one can expect SEEBIC signals to be uniquely capable of probing the equilibrium spatial distribution of electrons in materials. While beam deflection-based approaches (e.g., DPC)[25] have been used to image the charge density and associated electric fields in materials, these approaches report on the total charge density, including contributions of the atomic nuclei. Considering only secondary electrons generated through impact scattering (and not the long-ranged dipole scattering that gives rise to excitations over longer distances) the SEEBIC intensity can be roughly understood as a measure of the local density of electronic states that is modulated by the binding energy of the electrons relative to the Fermi level and the coupling strength between the initial and final electronic states due to the electric potential associated with the incident high-energy electrons. This position-resolved electron density information can be viewed as a partial Fourier complement to the momentum-resolved electronic band structure information that is now routinely studied with angle-resolved photoelectron spectroscopies.

Since the less tightly bound electrons residing in orbitals near the Fermi energy are those most readily liberated through the electron–electron scattering, they will be the principal contributors to the secondary electron production in the limit of low beam energies. As the energy of the incident electron is increased, however, deeper valence electrons and semicore states can also be excited with reasonable probability. Indeed, the majority of the contrast observed in SEEBIC images for typical STEM beam energies (>50 keV) is concentrated in the vicinity of the nuclei[7]. As is shown in this work, however, a careful analysis allows for contrast to also be resolved in the areas between the nuclei due to the scattering from bonding electrons.

Some of the authors have previously developed computational methods for evaluating transition probabilities between the bound electronic states of materials due to the potential associated with an idealized focused electron beam within the impulse approximation[26,27]. Here, we build on this technique to describe the scattering-induced excitations to a continuum of unbound free electron-like states. Inspired by prior TD-DFT studies of secondary electron generation[28,29] and related approaches developed for the simulation of angle-resolved photoemission spectra[30], we have adapted these time-dependent electronic structure theory methods to allow for the escape of electrons which become unbound as a result of the beam-like perturbation by imposing absorbing boundary conditions during the electronic dynamics. In the long simulation time limit, the change in the integrated electron density resulting from the perturbation provides a measure of ionization rate associated with applying the beam-like perturbation at a particular position, which can be used to simulate the spatial variation observed in the STEM-SEEBIC intensity. Notably, screening effects are naturally incorporated in this approach at the time-dependent self-consistent field level of theory. This is an important aspect of the current approach, as independent particle formalisms (which treat ionization as the removal of individual electrons from occupied orbitals) provide an unsatisfactory agreement to the observed spatial variation in secondary electron yields[22].

Real time time-dependent density functional theory simulations were performed with absorbing boundary conditions for an isolated cluster of 2H-phase WSe$_2$ constructed from previously published crystallographic data[31]. While small clusters of these semiconducting materials will exhibit significantly blue-shifted spectral features for transitions between bound excited states relative to the bulk materials due to quantum confinement[32], these effects are not expected to drastically influence the specific electronic structure information we seek in the current study (i.e., the position dependence of the probability for beam-induced ionization of valence electrons). All calculations were performed using a locally-modified version of the TD-DFT module[33,34] in the NWChem[35,36] electronic structure software, and employed the hybrid B3LYP[37–39] exchange-correlation functional and LANL2DZ[40] basis set and effective core potentials. The results are summarized in Fig. 4. Full details of our approach to simulating the beam-induced electronic excitation and secondary electron generation can be found the Supplementary Information Note 3.

Due to contributions to the overall ionization rate from dipole-allowed transitions to unbound and metastable, auto-ionizing states, the SEEBIC signal can be expected to show a significant delocalized component. For example, even for perturbations applied at the maximum distance away from the atomic centers within the material (i.e. where very little electron density resides) the calculated rate of

ionization is still nearly 70% that associated with the highest rate observed in our simulations. While this delocalized response is responsible for a relatively position-independent background contribution to the SEEBIC signal, sharply position-dependent features will be contributed by electronic transitions that are promoted through quadrupole and higher-order terms in the multipole expansion of the external Coulombic potential that are only active over short distances (i.e. through impact scattering). The ionization rate will also carry some position dependence for the dipole-allowed transition rates through modulations of the intensity of the electric field emanated by the external electron over the volume of the transition density, as even the electric dipole contribution to the matter-field interactions decays quadratically with distance from the field source. While the simulated valence ionization rates do exhibit a similar behavior to the SEEBIC images with respect to the variations in the SE yield for irradiation along the W–Se bonding and non-bonding directions, there are also notable discrepancies between the predicted (valence-only) yields and the total SE yields observed in the experiment. For instance, the experimental intensity ratio between the W and Se sites is not reproduced by our valence-only simulations. In contrast, the same computational method recovers the expected trend in total ionization yields across a series of small atomic test systems using all-electron TD-DFT calculations (see Supplementary Information Note 3). As such, we speculate that the discrepancy between the relative SEEBIC intensity contributed by the W and Se atoms and our simulated secondary electron yields arises (at least in part) due to the inability to capture ionization of core electrons in the current calculations, since effective core potentials have been used in the ab initio simulations as a matter of practical necessity in lieu of the 60/28 inner-shell electrons of the W/Se atoms. At the same time, these discrepancies may also suggest that to fully account for the observed intensity in the stacked heterostructures investigated in the experiments, simulations invoking an isolated cluster model of the embedded $WSe_2$ monolayer alone may be insufficient. This would be understandable, as the SEEBIC images were collected for $WSe_2$ laminated in multiple layers of materials with different dielectric properties, where even the atomistic details of the interfaces (leading to, e.g., spatial inhomogeneity in the SE escape depth) may nontrivially alter the SEEBIC intensity relative to the ionization rates of the isolated $WSe_2$ monolayer (see Supplementary Information Note 3 for a brief computational demonstration of this phenomenon).

Future efforts toward directly matching atomically resolved SEEBIC image intensities with theoretical ionization models will likely be significantly bolstered by using smaller, simpler experimental systems that are better matched to the tractability of the simulation. In the present case, the physical dimensions of the specimen (particularly the presence of multiple different lamination layers) preclude an in depth theoretical treatment. Therefore, simplified 2D systems, without the presence of lamination layers, that can be treated both experimentally and theoretically, are attractive future avenues for inquiry.

In this work, we have presented the case that the atomic contrast associated with a given lattice site in SEEBIC images can be understood as a summation of the total ionization cross sections for the inner-shell atomic orbitals, viewed as an image projection. We presented experimental SEEBIC data for a 2D heterostructure device designed to enable measurement of secondary electrons generated in a beam-sensitive, non-conducting $WSe_2$ monolayer. After careful processing to isolate the signal contributed by $WSe_2$, high resolution SEEBIC images present atomic scale features exhibiting discernable contrast differences between the different atomic species/sites. In the regions between the lattice sites, the same images also present heightened contrast along the W–Se bonding directions, which we hypothesized to originate from ionization of valence electrons in the interatomic bonding region due to impact scattering. To fully account for observed SEEBIC contrast variation and explain the origin of these more subtle effects,

improved theoretical approaches are necessary. To this end, we laid out a computational methodology based on TD-DFT that is equipped to explicitly capture the beam position dependence of ionization rates at the sub-atomic length scale. Simulated valence orbital ionization rates computed for an isolated cluster of $WSe_2$ support our conclusion that SEEBIC imaging can report on the spatial distribution of valence electrons in materials, along with the atomically-localized inner-shell electrons which contribute the lattice-level contrast that has been observed in this and other STEM-SEEBIC studies.

These results suggest that, in the future, direct imaging of the spatial distribution of electron orbital cross sections via SEEBIC may have the potential to reveal nuanced information about the local electron distribution and bonding. Coupled with techniques such as 4D STEM and EELS, which can both be acquired in parallel with SEEBIC, this technique promises to provide powerful insight into materials chemistry. The TD-DFT modeling approach presented here indicates the direction future investigations must take to fully account for SEEBIC contrast on the atomic scale. That the simplified model only partially accounts for the observed contrast in our experiments, suggests that this endeavor will be rich with new information.

## Methods

### Dry transfer stacking
The heterostructure device was assembled following a dry transfer protocol described below. First, single-layer $WSe_2$ and single-layer graphene and few-layer h-BN were mechanically exfoliated on 300 nm $SiO_2$/Si substrate with an adhesive tape. The transfer process was performed using a stacking station with a microscope and piezo controllers for sub-micron alignment. The layers were picked up using a clear polydimethylsiloxane (PDMS) stamp covered with a thin polycarbonate (PC) layer. Once all layers were on the stamp, the heterostructure was then released on the TEM chip by melting down the PC film with the proper alignment to ensure contact with the electrodes without shorting the device. The device was left overnight to avoid washing away the layers before washing off the film. The PC film was then removed in chloroform bath for 6 h followed by a few cycles of isopropanol wash and gentle blow dry with a nitrogen gun.

### STEM imaging
STEM imaging was performed using a Nion UltraSTEM 200 operated at an accelerating voltage of 100 kV and 80 kV as indicated with a nominal convergence angle of 30 mrad. Beam currents are listed with each image in the text. Electrical connections were made to the sample using a Protochips™ electrical contacting holder and the electrical leads were fed into a custom designed break-out box to facilitate making and breaking connections to the sample. A Femto DLPCA-200 transimpedance amplifier (TIA) was used for amplifying the SEEBIC signal. All SEEBIC images were acquired with a gain of $10^{11}$ V/A with a full band width of 1 kHz.

## Data availability
The data underlying Fig. 3 are available on Zenodo[41]. All raw data generated during the current study are available from the corresponding authors upon request.

## Code availability
The code to reproduce the analysis shown in Fig. 3 is available on GitHub and Zenodo[41].

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

## Acknowledgements

This work was supported by the U.S. Department of Energy, Office of Science, Basic Energy Sciences, Materials Sciences and Engineering Division (O.D. A.R.L., S.J.), and was performed at the Center for Nanophase Materials Sciences (CNMS), a U.S. Department of Energy, Office of Science User Facility. O.D. and S.J. were funded by DOE, SC, Basic Energy Sciences (BES) - ERKCK47. J.A. acknowledges the fund from support from the Army Research Office MURI (Ab Initio Solid-State Quantum Materials) Grant no. W911NF-18-1-043, from KACST-MIT Ibn Khaldun Fellowship for Saudi Arabian Women, and from Ibn Rushd Postdoctoral award from King Abdullah University of Science and Technology.

## Author contributions

O.D., J.A., A.R.L., D.E., and S.J. conceptualized the experiment, J.L.S. prepared the sample substrates, J.A. and B.H. prepared samples, O.D. collected data and performed data analysis, D.L. developed and implemented the theoretical treatment, A.R.L. and S.J. assisted with instrumentation, S.J. and D.E. supervised the project, O.D. and D.L. wrote the main manuscript draft and assembled figures, all authors assisted in editing, commenting, and clarifying the presentation. M.P.O. assisted with data analysis and manuscript editing.

## Competing interests

The authors declare no competing interests.
