## [Peer Review File · Nature Communications]

Direct Imaging of Electron Density with a Scanning Transmission Electron MicroscopeREVIEWER COMMENTS

Reviewer #1 (Remarks to the Author):

The authors report a study of the secondary electron EBIC signal measured in the aberration-corrected scanning transmission electron microscope, on a stacked 2D heterostructure (h-BN-encapsulated WSe₂) device. They also provide time-dependent DFT calculations to assist with the interpretation of their results. The experimental results are of high quality, as is the data processing for which the authors have a strong expertise. The difficulty of such experiments at atomic resolution is undeniable, from the making of a stacked 2D heterostructure device to the SEEBIC measurements and processing, and as such the work is very interesting. However, the lack of thorough interpretation of the contrast in the SEEBIC maps does not do any favour to the manuscript. Also, while the results undoubtedly show atomic-scale contrast, which in itself is quite an achievement, the fact that the SEEBIC signal might contain orbital information and contrast in the 2D maps can only remain a suggestion in the current state of the manuscript. As such, the current title and a statement in the abstract are not supported, contrary to the same points mentioned in the manuscript. Please find my comments below:

1. The title does not reflect the content of the manuscript. As it is now, the reader would expect to actually see electronic orbitals in the manuscript, which is not the case. In the abstract, the authors write "In turn, this suggests that subtle information about interlayer bonding and the effect on electron orbitals can be directly revealed with this technique". In the conclusion, the authors write "these results suggest that direct imaging of the spatial distribution of electron orbital cross sections via SEEBIC has the potential". While these sentences are more accurate descriptions of the results (i.e. the results show atomic contrast, but suggest orbital cross sections could be detected), they also underline that the current title is not a proper description of the work presented. Furthermore, existing works on orbital mapping using STEM-based techniques have previously been published (see comment below), and such broad title could suggest otherwise. Please modify the title, by making it more representative of the work.

For the same reasons, in the abstract, "Here, we apply the SEEBIC [...] to reveal the spatially resolved electron orbital ionization cross sections of an encapsulated WSe₂ layer." is not well supported by the work presented in the manuscript.

2. In this work, the authors provide a proof of principle of atomically-resolved SEEBIC, and suggest that it is a means to probe and detect electronic orbital signal at high resolution. This is very interesting and SEEBIC is certainly a valuable technique to pursue for this goal. Nevertheless, mapping electronic orbitals in STEM (or even TEM) has been performed before, with only few reports existing in the literature (e.g. TEM: Nature 401 (6748) (1999) 49–52, STEM: Ultramicroscopy 177, 26–29 (2017), Physical Review Letters 128, 116401 (2022), etc.). The manuscript would benefit from being put in the appropriate scientific context by acknowledging previously published work on orbital mapping using STEM-related techniques. Specifically, orbital mapping has been carried out using STEM-EELS and should be cited as such.

3. The SEEBIC maps in Figure 3 are only succinctly described in the text, despite being the core result of this study. It is quite disappointing that the spatial variations of the experimental SEEBIC contrast are not discussed in more depth, despite the discrepancies with simulations.

4. Why did the authors perform this experiment on h-BN-encapsulated WSe₂, and not on WSe₂ only (non-encapsulated)?

5. What is the impact of h-BN on the contrast in the SEEBIC maps? This point must be addressed to support the interpretation of atomic contrast in the SEEBIC maps, and the main conclusions. Specifically, how can the authors rule out the impact of h-BN on SEEBIC intensity observed in between W and Se atoms?

This is also important as the authors use this argument to explain why their simulation with an isolated cluster of WSe₂ is not sufficient (page 13): "This is understandable, as the SEEBIC images were collected for WSe₂ laminated in multiple layers of materials with different

dielectric properties, where the atomistic details of the interfaces nontrivially alter the SEEBIC intensity relative to the ionization rates of the isolated WSe₂ monolayer.”

6. With few-layer h-BN on top and below WSe₂, what is the escape depth of the SEEBIC signal measured in Figure 3?

7. The connection between the calculations in figure 4 and experiments in figure 3 should be made more obvious than it currently is. For instance, the authors claim (page 13) that “While the simulated valence ionization rates do exhibit position-dependent variations, there are notable discrepancies in the predicted rates and those observed in the SEEBIC images.”

Ideally, full simulations of the SEEBIC maps would be expected to support the claims in the manuscript and provide a more convincing interpretation of the origin of the contrasts. It is understandable that such simulations might not be possible due to the computing time necessary for TDDFT, therefore how would experimental current profiles (from figure 3g) compare to the ionization rate line profiles calculated in figure 4d, e?

8. Page 13: The TDDFT calculations demonstrate that the simplified model (isolated cluster of WSe₂) only partially accounts for the observed experimental contrast, and the comparison with experiments is not discussed in depth. Therefore, the experiments are not reproduced by simulations in the presented work, and the orbital contrast can only be an unsupported suggestion.

9. Similar to the previous comment, the suggestion about “subtle information about interlayer bonding” is not supported by the results presented

10. Page 7: “These layers were not deliberately aligned with each other or with the WSe₂, thus creating several periodic background signals.” Could the authors provide FFT or diffraction to better describe the stack misalignment?

This would also provide support to the sentence (page 8): “Because the tile selection has been aligned to the atomic positions of the WSe₂, any other features generated by the rest of the heterostructure stack (the h-BN and graphene) are smeared out into an average background.”.

11. Page 7: what is meant by “site symmetry” for the k-means clustering approach? In the present case, the projection along the c axis in the HAADF images means that both W and Se atoms have 3 equidistant neighboring atoms located at 120° from each other. How is “site symmetry” accounted for from such projection?

12. Further details on the simulations (approximations, minimal formalism for Time Dependence, etc.) should be provided either in supplementary materials or in the manuscript. What does “real time” mean in “Real time time-dependent density functional theory simulations”?

13. Complementary chemical analyses (core-loss EELS) of the stack would be useful. The authors actually mention this without adding complementary data (page 6): “Additionally, the h-BN can be detected using core-loss electron energy loss spectroscopic (EELS) imaging.”).

14. Page 8 “The images shown in Figure 3(g) have...” Should be replaced by “The intensity in the images shown in Figure 3(g) has...”

15. Add scale bar in optical micrograph of the stacked 3D heterostructure

Reviewer #2 (Remarks to the Author):

In this manuscript, Dyck, et al., performed SEEBIC of an encapsulated WSe₂ layer. The SEEBIC signals from W and 2Se atomic sites were quantified after averaging of clusters of a decent lattice-fringe image assisted by a good HAADF image. The experiments with atomic-scale contrast in the raw SEEBIC image and the data processing with deep learning are both well performed. However,

it is hard to understand how the authors came to the conclusion that the SEEBIC images show electron orbitals. I guess they got this conclusion from the fact that the average images in Fig. 3g show some distribution and that there is a finite intensity between atoms. If that is the case, they should have taken into account the large broadening effect from the finite incident probe, the low signal-to-noise and the averaging procedure. The image is a convolution between the physical quantity and the point spread function of the imaging system. If my understand is wrong, please clarify. Furthermore, the SE yield is usually considered to be from inner-shell excitation. The authors did not show how much of the SE yield is contributed from the bonding orbitals, as well as which orbitals, in the manuscript. From the semi-empirical formula (e.g., J. Appl. Phys. 54, R1 (1983)), the SE yield at a high beam energy from a single Se atom is smaller than that from a W atom, but WSe₂ has two Se atoms at one projected site. SE yield from two Se atoms at 20 keV is larger than a W atom, 0.56 vs 0.38. The authors should discuss the discrepancy between the empirical results with their first-principle calculations. Other questions that should be addressed are as follows,

1. The interlayer effect is not clear, since there is no comparison of SEEBIC signal from devices without BN layers and with BN layers. The surface status of the WSe₂ layer may also affect the SEEBIC signal and leads to discrepancies away from theory.
2. The statement in line 5 in page 10 should be corrected. There are a couple of works that have achieved atomic resolution by SE signal either using a SE detector or EBIC (ref. [7]).
3. There are nanoscale contrast fluctuations in Fig. 3b. What are the main origins? Additionally, there are huge variations of the SEEBIC intensity over different clusters from both W and Se-Se. And if the huge variation of the SEEBIC signal is due to the measurement uncertainty, then the difference of SEEBIC between W and Se-Se is not statistically significant. The authors should give more discussions.
4. h-BN layer is not conducting. What is the main motivation to add two h-BN layers in the device? Can we see better SEEBIC signal if there is no h-BN layer between WSe₂ and graphene?

In case of these concerns, the current version of the manuscript is not recommended to publish anywhere.

Reviewer #3 (Remarks to the Author):

The paper presents interesting experimental data on atomic resolution SEEBIC imaging on 2D materials and notes that the local contrast, obtained after substantial data treatment and AI based template matching like techniques shows an atomic structure that is related to the local electron density and could reveal bonding and orbital information.

The experimental setup is very interesting and clearly state of the art. The data processing is also quite involved due to the very low signal to noise ratio. The fact that such high reliance on data treatment is needed to gain any signal at all should be taken as a warning for over interpretation, but the results are at least plausible.

The paper then continues with the interpretation of this data and this is where I start to disagree or would at least like to see more proof for the claims given.

To be more specific:

The authors claim on p10 "and the low-loss (valence) EELS signal reports on the excitation of delocalized bosonic quasiparticles and collective electronic modes; one can expect SEEBIC signals to be uniquely capable of probing the equilibrium spatial distribution of electrons in materials."

I don't understand the distinction of excitations with primary fast electrons and the outer shell electrons which is common to both SEEBIC and low loss EELS to lead to a fundamentally different signal. In fact I would have thought that low loss EELS would be an excellent signal to compare the SEEBIC with as this would also show atomic resolution details which have been demonstrated to be rather hard to interpret with respect to e.g. preservation of elastic contrast in combination with delocalised excitations and partial coherence, which should also affect SEEBIC for the above reasons (same excitations taking place because stimulus is the same). I suggest the authors to add such low loss data and show e.g. the integrated intensities in an energy window where also SE

emission and capturing is expected in the SEEBIC setup. Intuitively, I would expect an image integrated in the say 5-20eV region to be very similar to the SEEBIC image obtained via datamining but now with vastly better signal to noise ratio and probably not requiring the datatreatment nor the very slow scanning at high currents.

This is an important key point of the paper. The authors claim that SEEBIC somehow has a different nature of scattering and therefore uniquely brings out orbital electronic density information. In my point of view there is absolutely no difference in the scattering mechanism between SEEBIC and e.g. low loss EELS, both originate from the same interaction of the fast electron and the electrons in the material. The only difference is in what happens at the detection side (SE electrons vs energy loss of fast electrons) and how this is affected by e.g. reabsorption of the SEs but this aspect is almost completely ignored in the current paper.

In summary I think the data is interesting but the interpretation makes unfounded assumptions and creates expectations that are not backed up with plausible evidence. If the simulations are right, they would equally affect a low loss EELS signal and hence deliver a signal that we already had for quite some time now, but which has often turned out to be very hard to interpret exactly in that regime where many possible scattering mechanisms compete.

A reworked version of the paper addressing this point more carefully and with experimental proof would significantly improve the scientific value of the experimental findings.

We appreciate the time and effort the reviewers have clearly undertaken to produce their thorough critique of our manuscript. Through their comments we have made significant changes to the manuscript especially regarding the clarification of what is being claimed and what is not claimed. Their comments have greatly improved the presentation.

We have changed the title, added significant supplemental materials, included new theoretical input, added as an author an expert who contributed to these changes.

Detailed replies to each comment follow.

Reviewer #1 (Remarks to the Author):

The authors report a study of the secondary electron EBIC signal measured in the aberration-corrected scanning transmission electron microscope, on a stacked 2D heterostructure (h-BN-encapsulated WSe₂) device. They also provide time-dependent DFT calculations to assist with the interpretation of their results. The experimental results are of high quality, as is the data processing for which the authors have a strong expertise. The difficulty of such experiments at atomic resolution is undeniable, from the making of a stacked 2D heterostructure device to the SEEBIC measurements and processing, and as such the work is very interesting. However, the lack of thorough interpretation of the contrast in the SEEBIC maps does not do any favour to the manuscript. Also, while the results undoubtedly show atomic-scale contrast, which in itself is quite an achievement, the fact that the SEEBIC signal might contain orbital information and contrast in the 2D maps can only remain a suggestion in the current state of the manuscript. As such, the current title and a statement in the abstract are not supported, contrary to the same points mentioned in the manuscript. Please find my comments below:

We appreciate the reviewer's recognition of the experimental difficulty and quality of the results presented. Indeed, it was not obvious at the outset that the results presented would be physically possible. The reviewer mentions that we have clearly shown atomic scale contrast in the SEEBIC images but that orbital information remains merely a suggestion. It is on this point that we have not been clear enough. SEEBIC imaging does not detect nuclear scattering, as is the case in HAADF. The atomic-scale contrast observed in SEEBIC comes from interaction with the sample electrons through ionization. Thus, we believe the atomic-scale contrast observed *is* the detection of electron orbitals. We must be careful to specify that we cannot distinguish between *different* orbitals but the contrast that is observed is essentially the projected

ionization probability of all orbitals, summed together. We have now made extensive clarifications this in the manuscript to reflect this as well as a revision in the wording of the title.

1. The title does not reflect the content of the manuscript. As it is now, the reader would expect to actually see electronic orbitals in the manuscript, which is not the case. In the abstract, the authors write “In turn, this suggests that subtle information about interlayer bonding and the effect on electron orbitals can be directly revealed with this technique”. In the conclusion, the authors write “these results suggest that direct imaging of the spatial distribution of electron orbital cross sections via SEEBIC has the potential”. While these sentences are more accurate descriptions of the results (i.e. the results show atomic contrast, but suggest orbital cross sections could be detected), they also underline that the current title is not a proper description of the work presented. Furthermore, existing works on orbital mapping using STEM-based techniques have previously been published (see comment below), and such broad title could suggest otherwise.

Please modify the title, by making it more representative of the work.

The authors apologize for the confusion surrounding the title of our manuscript. It was not our intent to imply that the electronic orbitals can be visualized individually with the current technique, but rather that the occupied electronic orbitals can be visualized collectively. We have modified the title, and our discussion in the text around the visualization of the equilibrium spatial distribution of electrons in materials accordingly.

However, our data *directly* shows electron orbital/electron density contrast (from all orbitals together). In the future, this orbital contrast might be leveraged to reveal changes in e.g. interlayer bonding. This distinction has now been made clearer. The reader can see the total projected electron orbitals in Fig. 3(g). The discussion surrounding this figure has been extended to make this clear.

For the same reasons, in the abstract, “Here, we apply the SEEBIC [...] to reveal the spatially resolved electron orbital ionization cross sections of an encapsulated WSe₂ layer.” is not well supported by the work presented in the manuscript.

We believe this claim is supported by Fig. 3(g). We have made extensive revisions to help clarify the significance of this image and its interpretation.

2. In this work, the authors provide a proof of principle of atomically-resolved SEEBIC, and suggest that it is a means to probe and detect electronic orbital signal at high resolution. This is very interesting and SEEBIC is certainly a valuable technique to pursue for this goal. Nevertheless, mapping electronic orbitals in STEM (or even TEM) has been performed before, with only few reports existing in the literature (e.g. TEM: Nature 401 (6748) (1999) 49–52, STEM: Ultramicroscopy 177, 26-29 (2017), Physical Review Letters 128, 116401 (2022), etc.). The manuscript would benefit from being put in the appropriate scientific context by acknowledging previously published work on orbital mapping using STEM-related techniques. Specifically, orbital mapping has been carried out using STEM-EELS and should be cited as such.

We have added further references to the literature regarding the use of STEM to visualize electron orbitals, as recommended by the referee, and have provided a brief discussion. In addition, we believe that we have addressed this issue raised by the reviewer by clarifying the electron spatial distribution mapping afforded under the SEEBIC techniques. We note in passing, also, that techniques for mapping charge densities in materials that rely on beam deflection (e.g., DPC) are not able to isolate the contributions from electronic degrees of freedom, as the deflection of primary electrons is also influenced by the significant positive charge contributed by nuclei. In this regard, SEEBIC is a unique capability for exclusively mapping the equilibrium spatial distribution of the electronic (i.e., not total=electronic+nuclear) charge densities in materials.

3. The SEEBIC maps in Figure 3 are only succinctly described in the text, despite being the core result of this study. It is quite disappointing that the spatial variations of the experimental SEEBIC contrast are not discussed in more depth, despite the discrepancies with simulations.

The reviewer is clearly correct in this assessment, as we failed to properly communicate the significance we attribute to, particularly, Fig. 3(g). The discussion has been significantly enhanced, expanded, and clarified.

4. Why did the authors perform this experiment on h-BN-encapsulated WSe₂, and not on Wse₂ only (non-encapsulated)?

Non-encapsulated Wse₂ suffers from beam damage under the imaging conditions used here. The main purpose of the h-BN layers for this experiment were to mechanically stabilize the Wse₂ layer to make it more robust against beam damage. Even with this protective layer, beam damage was eventually observed

and we faced great difficulties obtaining damage-free images with sufficiently long exposure times to generate an acceptable SEEBIC signal. Future experimental design improvements aim to address this situation.

5. What is the impact of h-BN on the contrast in the SEEBIC maps? This point must be addressed to support the interpretation of atomic contrast in the SEEBIC maps, and the main conclusions. Specifically, how can the authors rule out the impact of h-BN on SEEBIC intensity observed in between W and Se atoms?

This is also important as the authors use this argument to explain why their simulation with an isolated cluster of WSe₂ is not sufficient (page 13): “This is understandable, as the SEEBIC images were collected for WSe₂ laminated in multiple layers of materials with different dielectric properties, where the atomistic details of the interfaces nontrivially alter the SEEBIC intensity relative to the ionization rates of the isolated WSe₂ monolayer.”

There are two main comments on this issue: 1) by aligning the tiles to the atom positions observed in the HAADF image we are specifically selecting the WSe₂ response instead of the other materials in the stack. In this way, the other materials generate a background signal on top of which the WSe₂ signal emerges. Because the tiles are not spatially aligned with these other signals and because we have a large number of sample tiles/images (>700) these signals are smeared out. 2) Nevertheless, we cannot assume that the WSe₂ signal is not affected by the h-BN encapsulation layers. For example, the Se atoms are in much closer proximity to the h-BN layers than the W atoms and are likely responsible for most of the electronic interactions between the two materials. This may affect the mean orbital response of the Se atoms in comparison to the response of bare WSe₂. It is this reasoning that prompts us to infer that “direct imaging of the spatial distribution of electron orbital cross sections via SEEBIC has the potential to reveal nuanced information about the local electron distribution and bonding.” We are not able to conclusively demonstrate this, but since we are able to directly image the total electron orbital scattering cross sections it serves to reason that such capabilities are within the realm of possibility.

6. With few-layer h-BN on top and below WSe₂, what is the escape depth of the SEEBIC signal measured in Figure 3?

At present, there is significant uncertainty in the precise value of the escape depth. Escape depth is often treated as a constant, however for bulk materials an exponential decay is usually accepted as the more

accurate model. Nevertheless, theoretical treatments typically approach the problem from the perspective of continuous (non-atomistic) media. Clearly, at the scale in which we are operating here, the assumption of continuous media is no longer appropriate. Thus, the concept of escape depth becomes ill-defined as it depends on emission position, emission direction, and crystallographic orientation of the overlayers. These factors have now been highlighted in the discussion.

7. The connection between the calculations in figure 4 and experiments in figure 3 should be made more obvious than it currently is. For instance, the authors claim (page 13) that “While the simulated valence ionization rates do exhibit position-dependent variations, there are notable discrepancies in the predicted rates and those observed in the SEEBIC images.”

Ideally, full simulations of the SEEBIC maps would be expected to support the claims in the manuscript and provide a more convincing interpretation of the origin of the contrasts. It is understandable that such simulations might not be possible due to the computing time necessary for TDDFT, therefore how would experimental current profiles (from figure 3g) compare to the ionization rate line profiles calculated in figure 4d, e?

We have now significantly expanded the discussion fig 3 and clarified the purpose of the simulations, which was not to establish a one-to-one correspondence with the experimental data but to illustrate the direction one must take to approach the problem. We also now present TDDFT results showing the full simulated image in the revised manuscript, to aid the comparison to the experimental result as this reviewer suggests.

8. Page 13: The TDDFT calculations demonstrate that the simplified model (isolated cluster of WSe_2) only partially accounts for the observed experimental contrast, and the comparison with experiments is not discussed in depth. Therefore, the experiments are not reproduced by simulations in the presented work, and the orbital contrast can only be an unsupported suggestion.

We do not rely on the theoretical treatment to support the claim that electron orbital contrast is being observed. This claim is demonstrated by the experimental results, fig 3g. It is, rather, these experimental results that motivate a fresh look at the treatment of secondary electron emission (SE) at the atomic scale (ionization). Established first principles descriptions of SE emission do not consider the atomic nature of materials and therefore cannot describe, for example, the different contrast exhibited by the W and Se-Se lattice sites. The purpose of the TDDFT calculations is to provide a first step in this direction, laying out specifically the strategy employed and initial results which *can* account for different contrast levels at different lattice sites. A one-to-one match between experiment and theory is not tractable with currently

available electronic structure methods for this particular experiment. Nevertheless, it is clear that this is the direction required to fully account for the observed contrast, which is not obvious when examining lower resolution SEEBIC images. The presentation has been substantially revised to make these points clear.

9. Similar to the previous comment, the suggestion about “subtle information about interlayer bonding” is not supported by the results presented

This statement is, to some degree, speculative. We claim that these results “suggest” that information about interlayer bonding can be revealed. We do not intend to claim that we have revealed it here. In light of our clarifications elsewhere, especially the expanded discussion regarding Fig. 3(g), we believe the reviewer will now agree with the statement. It now reads: “In turn, this suggests that, **in the future**, subtle information about interlayer bonding and the effect on electron orbitals **could** be directly revealed with this technique.”

10. Page 7: “These layers were not deliberately aligned with each other or with the Wse2, thus creating several periodic background signals.” Could the authors provide FFT or diffraction to better describe the stack misalignment?

This would also provide support to the sentence (page 8): “Because the tile selection has been aligned to the atomic positions of the Wse2, any other features generated by the rest of the heterostructure stack (the h-BN and graphene) are smeared out into an average background.”.

An image of the misalignment is now provided in the supplemental information.

11. Page 7: what is meant by “site symmetry” for the k-means clustering approach? In the present case, the projection along the c axis in the HAADF images means that both W and Se atoms have 3 equidistant neighboring atoms located at 120° from each other. How is “site symmetry” accounted for from such projection?

This is, indeed, poorly chosen wording. Here, we mean that the W and Se-Se lattice sites appear *rotated* with respect to each other. Even without intensity information at a particular lattice site, one can determine whether it ‘should’ be a W or Se-Se site based on the position of the neighboring atoms. This has been clarified in the text. Instead of “sight symmetry” we now refer to it as “rotational symmetry”.

12. Further details on the simulations (approximations, minimal formalism for Time Dependence, etc.)

should be provided either in supplementary materials or in the manuscript. What does “real time” mean in “Real time time-dependent density functional theory simulations”?

A document containing supporting information that gives the details of the time-domain electronic dynamics simulations has been provided along with the revised manuscript.

13. Complementary chemical analyses (core-loss EELS) of the stack would be useful. The authors actually mention this without adding complementary data (page 6): “Additionally, the h-BN can be detected using core-loss electron energy loss spectroscopic (EELS) imaging.”).

Core-loss EELS data is now included in the supplemental information.

14. Page 8 “The images shown in Figure 3(g) have...” Should be replaced by “The intensity in the images shown in Figure 3(g) has...”

We agree with the reviewer, and have made the requested revision.

15. Add scale bar in optical micrograph of the stacked 3D heterostructure

We agree with the reviewer, and have made the requested revision.

Reviewer #2 (Remarks to the Author):

In this manuscript, Dyck, et al., performed SEEBIC of an encapsulated WSe₂ layer. The SEEBIC signals from W and 2Se atomic sites were quantified after averaging of clusters of a decent lattice-fringe image assisted by a good HAADF image. The experiments with atomic-scale contrast in the raw SEEBIC image and the data processing with deep learning are both well performed. However, it is hard to understand how the authors came to the conclusion that the SEEBIC images show electron orbitals. I guess they got this conclusion from the fact that the average images in Fig. 3g show some distribution and that there is a finite intensity between atoms. If that is the case, they should have taken into account the large broadening effect from the finite incident probe, the low signal-to-noise and the averaging procedure. The image is a

convolution between the physical quantity and the point spread function of the imaging system. If my understand is wrong, please clarify. Furthermore, the SE yield is usually considered to be from inner-shell excitation. The authors did not show how much of the SE yield is contributed from the bonding orbitals, as well as which orbitals, in the manuscript. From the semi-empirical formula (e.g., J. Appl. Phys. 54, R1 (1983)), the SE yield at a high beam energy from a single Se atom is smaller than that from a W atom, but WSe₂ has two Se atoms at one projected site. SE yield from two Se atoms at 20 keV is larger than a W atom, 0.56 vs 0.38. The authors should discuss the discrepancy between the empirical results with their first-principle calculations. Other questions that should be addressed are as follows,

Using the standard first principles electronic structure methods available in the area of computational condensed matter physics, core electrons are ‘pseudo-ized’, and treated by an effective core potential rather than being included in the self-consistent field equations with the valence electrons. As such, these simulations are not suitable for the calculation of core ionization. It is quite possible that the neglect of core electron contributions to the secondary electron yields is responsible for the deviation from the differences in absolute SE yields for irradiation at the W and Se sites, but we are unable to confirm this due to the lack of suitable methods to treat the core electrons under the canonical electronic structure theoretical methods. In lieu of this, we have provided (in the supplemental information document supplied with our revised manuscript) TD-DFT data for a series of closed shell atomic species of lower atomic number, for which conventional (i.e., non-relativistic) electronic structure theory calculations can be performed under our approach in an all-electron framework. We find that the trend in SE yield due to electron impact from our simplified treatment is in satisfactory qualitative agreement with the analogous trend in the total ionization cross sections. This lends support for our claims that the TD-DFT approach used in the current study is capable of providing meaningful insights into the position-dependent yield of valence secondary electrons generated by idealized electron beam-like external potentials. This is not to say that all of the relevant physics represented in the semi-empirical formula cited by this reviewer is captured under the current approach, and we have given further details of the approximations we make in our computational methods as well their implications for comparing to experiment in the SI document.

As only the valence electrons can be simulated for the heavier elements comprising the material of interest in this study, all of the SE yield in our calculation originates from emission of electrons from the valence orbitals. This has been clarified in the revised manuscript. The authors’ agree that the majority of atomic contrast in SEEBIC is contributed by the ionization of inner-shell electrons. However, even though ionization of electrons closer to the Fermi energy may not contribute the majority of the atomic contrast, it is not the case that valence electrons are less likely to be ejected from the material at a given beam energy

than the more tightly bound inner-shell electrons. Indeed, for the case of isolated atoms the total/integrated ionization cross sections calculated within the binary encounter dipole formalism (which reproduces the experimentally measured cross sections remarkably well) are orders of magnitude larger for electrons residing in valence atomic orbitals than for the core electrons.

Finally, we note that the specific and limited intent of the simulations in this study is to demonstrate the spatial variation in the rate of beam-induced ejection of valence electrons in the W-Se bonding region. We have revised the text to better clarify the scope of the simulations.

We agree that the image is a convolution between the physical quantity and the point spread function of the imaging system. The primary purpose of the paper is to ask (answer) what, exactly, is the physical quantity we are measuring. With HAADF imaging we are measuring, primarily, nuclear scattering. The HAADF image is a convolution between the nuclear scattering cross section of the sample and the point spread function of the instrument. So, with HAADF imaging we can say we are directly imaging nuclear scattering cross sections (or projections of them). Nuclear scattering is invisible to SEEBIC imaging, nevertheless we can acquire an “atomically resolved” SEEBIC image that shows contrast from one lattice site to another. This cannot be explained with the standard approach used to describe SE yield (e.g. J. Appl. Phys. 54, R1 (1983)) because individual atoms within a material are not treated individually. To address this problem we must recourse to a description of ionization of single atoms. Once it is recognized that the entirety of the atomic contrast in SEEBIC is due to ionization, one is forced into the conclusion that the SEEBIC image is a projection of the total electron orbital ionization cross section. Individual orbitals are not resolved or distinguished from one another. These points have been made more clearly in the text.

1. The interlayer effect is not clear, since there is no comparison of SEEBIC signal from devices without BN layers and with BN layers. The surface status of the WSe₂ layer may also affect the SEEBIC signal and leads to discrepancies away from theory.

We have revised the wording in the manuscript to make it clearer that we do not observe an unambiguous interlayer effect here, rather, interlayer effects might be measurable using this technique.

2. The statement in line 5 in page 10 should be corrected. There are a couple of works that have achieved atomic resolution by SE signal either using a SE detector or EBIC (ref. [7]).

We apologize for the lack of clarity in our presentation. We intended to convey that there have been previous theoretical efforts to study SE emission from a theoretical perspective at the macro-scale, but that those approaches either only consider the contributions to the SE yield from inner-shell electrons (and not the valence electrons which contribute the signal of interest in the current works), or are based on bulk-defined quantities and are not immediately applicable for understanding the spatial variations in the SE yield on atomic length scales. Our presentation has been substantially revised in many locations to help clarify these points.

3. There are nanoscale contrast fluctuations in Fig. 3b. What are the main origins? Additionally, there are huge variations of the SEEBIC intensity over different clusters from both W and Se-Se. And if the huge variation of the SEEBIC signal is due to the measurement uncertainty, then the difference of SEEBIC between W and Se-Se is not statistically significant. The authors should give more discussions.

Contrast fluctuations observed in Fig. 3b are most likely due to layer variations in the h-BN which gradually exhibits evidence of beam damage. The issue of contrast variability in the SEEBIC intensity between W and Se-Se lattice sites is an interesting discussion and worth addressing more carefully. From the SEEBIC data by itself we cannot unambiguously assign labels to the W and Se-Se lattice sites for the reason the reviewer mentions--the contrast variability is too high. Moreover, we did not find it possible to unambiguously assign atom positions based on the SEEBIC data either. This is the reason we label the analytic workflow with "No Info." for the SEEBIC data in Fig. 3c. To address this challenge we rely on the simultaneously acquired HAADF image which unambiguously shows lattice positions and atomic contrast between W and Se-Se lattice sites. With this information we can determine both lattice positions and type for *both* images. We can then average the observed response from the SEEBIC data to increase our SNR. The uncertainty here is the uncertainty of the *mean* sampled from >700 datapoints which is much smaller than the uncertainty for each datapoint. Fig. 3e has now been updated with uncertainty intervals representing the uncertainty in the calculation of this mean. Likewise, the manuscript text has been updated to elaborate these details directly.

4. h-BN layer is not conducting. What is the main motivation to add two h-BN layers in the device? Can we see better SEEBIC signal if there is no h-BN layer between WSe₂ and graphene?

The main purpose for the h-BN is to form mechanically stabilizing encapsulation layers. The bare WSe₂ material is easily damaged by e-beam irradiation and will not withstand the dose required for SEEBIC imaging. We speculate that use of only graphene encapsulation, as suggested by the reviewer, may improve both the SEEBIC signal and robustness against radiation damage. Future experiments along these lines are planned, the results of which will be presented elsewhere.

In case of these concerns, the current version of the manuscript is not recommended to publish anywhere.

It is unfortunate that the reviewer comes to this conclusion, but we hope the substantial clarifications and improvements to the manuscript will convince the reviewer otherwise.

Reviewer #3 (Remarks to the Author):

The paper presents interesting experimental data on atomic resolution SEEBIC imaging on 2D materials and notes that the local contrast, obtained after substantial data treatment and AI based template matching like techniques shows an atomic structure that is related to the local electron density and could reveal bonding and orbital information.

The experimental setup is very interesting and clearly state of the art. The data processing is also quite involved due to the very low signal to noise ratio. The fact that such high reliance on data treatment is needed to gain any signal at all should be taken as a warning for over interpretation, but the results are at least plausible.

The paper then continues with the interpretation of this data and this is where I start to disagree or would at least like to see more proof for the claims given.

To be more specific:

The authors claim on p10 "and the low-loss (valence) EELS signal reports on the excitation of delocalized bosonic quasiparticles and collective electronic modes; one can expect SEEBIC signals to be uniquely capable of probing the equilibrium spatial distribution of electrons in materials."

I don't understand the distinction of excitations with primary fast electrons and the outer shell electrons which is common to both SEEBIC and low loss EELS to lead to a fundamentally different signal. In fact I would have thought that low loss EELS would be an excellent signal to compare the SEEBIC with as this would also show atomic resolution details which have been demonstrated to be rather hard to interpret with

respect to e.g. preservation of elastic contrast in combination with delocalised excitations and partial coherence, which should also affect SEEBIC for the above reasons (same excitations taking place because stimulus is the same). I suggest the authors to add such low loss data and show e.g. the integrated intensities in an energy window where also SE emission and capturing is expected in the SEEBIC setup. Intuitively, I would expect an image integrated in the say 5-20eV region to be very similar to the SEEBIC image obtained via datamining but now with vastly better signal to noise ratio and probably not requiring the datatreatment nor the very slow scanning at high currents.

Different nature of detecting ionization with EELS and SEEBIC:

An important aspect of the SEEBIC detection is that it represents an absolute measurement of the rate of holes produced in the specimen through SE emission. If the goal is to reproduce the SEEBIC by integration of the EELS spectrum, one would need to (1) somehow disentangle the contribution to the measured loss function from ionization and excitations between bound states in the spectrum, and sum only over the former and (2) collect and energy-analyze 100% of the scattered primary electrons. For EELS collected using standard size detectors in the “on-axis” geometry, only the small (in-plane) momentum transferring scattering events are registered, which will correspond to an incomplete set of electronic excitations in the material with more stringent selection rules relative to the total ionization rate measured with SEEBIC. While these higher-angle scattering events may be significantly rarer than the low-angle ones, they are intuitively the ones more likely to transfer sufficient kinetic energy to result in ionization, and therefore shouldn't be regarded as negligible in this context.

Portions of the SEEBIC signal are undoubtedly represented in the low-loss EELS signal but what makes SEEBIC unique in this regard is that it only detects ionization to the exclusion of all other signals. The integration of a 5-20 eV range in the low-loss signal would, for most materials, include a large bulk plasmon contribution which is not due to ionization and is typically delocalized. This does open an interesting avenue for a quantitative comparison of the two detection methods, that we hope to explore in future work.

We now directly address these discussion points in the manuscript introduction and in the discussion surrounding figure 3.

Data mining

The data mining procedure is, admittedly, rather involved. However, what it is actually doing is very straight forward. The challenge is in telling the computer what to do. In this case, all that we are doing is cutting

out an image tile around every atom position and averaging all the tiles from the W lattice sites and all the tiles from the Se-Se lattice sites. This is a very straight forward *idea* for a human; very difficult to translate to the computer. In our case, we already had in place much of the code to perform this analysis, so it was not developed from scratch. Likewise, we provide the commented code to the reader on github so neither is the reader forced to start from scratch. We hope that this makes the task of examination and replication much easier and will be useful to the scientific community.

The data mining procedure is also performing an additional service. Because the tiles are being aligned with the WSe₂ lattice sites, the signal from the WSe₂ gets summed up but other signals from the h-BN and graphene get smeared out into a background signal. If we were able to get a strong EELS signal that represented just the SE emission, we would still need to perform some procedure like this to discriminate the layer we are interested in.

The manuscript text has been revised in light of these points.

This is an important key point of the paper. The authors claim that SEEBIC somehow has a different nature of scattering and therefore uniquely brings out orbital electronic density information. In my point of view there is absolutely no difference in the scattering mechanism between SEEBIC and e.g. low loss EELS, both originate from the same interaction of the fast electron and the electrons in the material. The only difference is in what happens at the detection side (SE electrons vs energy loss of fast electrons) and how this is affected by e.g. reabsorption of the SEs but this aspect is almost completely ignored in the current paper.

We agree that the SEEBIC signal should exist in the EELS signal. There is no way to induce SE emission without energy loss to the primary electrons. The advantage of SEEBIC is that this is the *only* signal whereas there are a mixture of many signals that are difficult to interpret in EELS. As an example, why do people bother acquiring HAADF images when, in principle, all of this information, and more, is contained in the core-loss EELS data?

These points are now directly discussed.

In summary I think the data is interesting but the interpretation makes unfounded assumptions and creates expectations that are not backed up with plausible evidence. If the simulations are right, they would equally affect a low loss EELS signal and hence deliver a signal that we already had for quite some time now, but

which has often turned out to be very hard to interpret exactly in that regime where many possible scattering mechanisms compete.

It is precisely this difficulty in interpretation that the SEEBIC image addresses. These points have now been added to the manuscript and we thank the reviewer for weighing in with this perspective. It has improved the presentation and scientific depth of the manuscript.

A reworked version of the paper addressing this point more carefully and with experimental proof would significantly improve the scientific value of the experimental findings.

We hope our improvements and discussion in the reworked manuscript meet with the reviewer's approval and thank them very much for their constructive critique.

REVIEWER COMMENTS

Reviewer #1 (Remarks to the Author):

I appreciate the efforts from the authors to explain their points of view and to take into account all comments formulated during the referral process. The authors clarified most points of concern arising from my previous remarks, and the changes made to the manuscript are helpful. In my view, the revised version of the manuscript has been improved and carries a clearer message, of interest to the electron microscopy community and beyond. However, the fact that TDDFT simulations hardly reproduce the experimental data, even qualitatively, is disappointing, especially since it does not support the interpretation of the atomically-resolved SEEBIC contrast in terms of total electron orbital cross-section. Nevertheless, the explanations provided by the authors in their responses, and the more suggestive tone employed in the revised manuscript help clarify the position of their work. I do have additional comments:

- Figure 3(g), which is central to the discussion and the main message of this manuscript, as underlined by the authors in their responses and revisions. The histograms in Figure 3(e), which have been modified in this revised version of the manuscript, are directly linked to the intensity range of the SEEBIC maps in Figure 3(g). Considering the histograms, how is the contrast in the SEEBIC maps in Figure 3(g) affected by the current range? i.e. could the authors provide SEEBIC maps for currents in various ranges, e.g. above 0.2 pA and up to 0.8 pA. What is the meaning of having SEEBIC current below 0.0 pA (after the rescaling)?

The reason for choosing an intensity range between 0.0 and 0.2 pA should be indicated. Providing a clear demonstration for the choice of current limits in the SEEBIC maps would help readers assess the robustness of the interpretation.

- In their additions to the revised manuscript, the authors indicate that (p. 12) "This signal is also present in EELS data, which detects energy losses to the primary beam electrons, however, many other energetic transitions are simultaneously detected with EELS. This makes the interpretation of the EELS signal more difficult."

Could the authors clarify what they mean by "many other energetic transitions are simultaneously detected with EELS", and "This makes the interpretation of the EELS signal more difficult."? For the latter, individual excitations in EELS can typically be discriminated as a function of energy-loss and interpreted individually.

Reviewer #2 (Remarks to the Author):

In this revised manuscript, the authors have addressed most concerns and significantly improved the readability. But a key concern still exists, that is, what exactly is the origin of the atomic contrast in the SEEBIC image? In the manuscript, the authors tried to emphasize that the images do not show distinct electron orbitals, then why the concept of orbitals should be introduced? I agree with the authors that the SEEBIC image contrast comes from the electronic charge density excluding nuclear charge density. So good wording for the title may be the electron charge density or ionization cross section. Words like "Total electron orbital" or "summed electron orbitals" are weird and not well-defined. In electron microscopy especially, "orbital" is usually used when different orbitals can be distinguished such as d-orbitals from core-loss EELS. Using orbital in the work is very misleading and seems over claimed.

Reviewer #3 (Remarks to the Author):

The revised paper by Dyck et al. has addressed some of the concerns raised by the reviewers. As all reviewers objected against some form of overselling the orbital imaging aspect in this otherwise remarkable experiment, I find it difficult to accept that the current paper still has a title which claims

"Imaging of Total Electron Orbital Cross Sections".

for the following reasons:

1st: Orbitals are non observable objects and can therefore not be imaged in experiments.
2nd: we don't see the total electron scattering cross section either as it discussed now better how e.g. plasmon excitations don't contribute to the SEEBIC contrast while they would occur in EELS.
3rd: I don't agree that the positive core of the atoms plays no role as it is this core that gives rise to the presence of all the states and their energies and determines the ionisation cross section. The fact that SE yield is sensitive to the work function further proves this point. (also the TDFT approach would not work if the cores wouldn't be present)

A more accurate title proposal could be

"Atomic scale mapping of local ionisation cross section reveals anisotropic local density of electronic states in 2D materials"

I also don't agree with the statement on line 82 that we get a direct imaging of the 'electronic structure'. In the conventional meaning of electronic structure, the band structure involves both reciprocal or real space position dependent information and the energy of bands. Here we get only the location and something related to density of states with the energy axis integrated out.

While line 234 more correctly claims that "The SEEBIC images shown in Figure 3(g) represent the total ionization cross section of the material, convolved with the point spread function of the instrument."

On line 237 another attempt is made to redefine what is being measured.

"Thus, the SEEBIC image represents the total projected electron orbital cross section of the specimen--all electron orbitals summed together and projected".

It is however unclear on what we project here and what the difference is between an orbital cross section and the ionisation cross section. And also why this is suddenly related to the sum (weighted somehow?) of the electron orbitals (or their density?).

It is remarkable to note that from about line 321 the paper suddenly becomes much more precise in describing these excitation processes which indicates that within the current author list there is ample experience and knowledge to significantly improve the loose statements made in the earlier paragraphs and to come up with a more precise title and descriptive text that does justice to this beautiful work, both experimentally and in terms of simulations.

REVIEWER COMMENTS

Reviewer #1 (Remarks to the Author):

I appreciate the efforts from the authors to explain their points of view and to take into account all comments formulated during the referral process. The authors clarified most points of concern arising from my previous remarks, and the changes made to the manuscript are helpful. In my view, the revised version of the manuscript has been improved and carries a clearer message, of interest to the electron microscopy community and beyond. However, the fact that TDDFT simulations hardly reproduce the experimental data, even qualitatively, is disappointing, especially since it does not support the interpretation of the atomically-resolved SEEBIC contrast in terms of total electron orbital cross-section.

We agree that it is unfortunate that the current TDDFT approach cannot be used to produce total ionization cross sections for the WSe₂ system, but this is ultimately a limitation of the solid state electronic structure theory capabilities of the quantum chemistry software used in the current work, which relies on pseudopotentials to describe inner-shell electrons. We made efforts to clarify the point in the revised manuscript that the current calculations are capable only of capturing the *valence* contribution to the SEEBIC, but that this would be the relevant contribution of the total cross section for understanding the SEEBIC intensity contributed in the W-Se bonding region by ionization of bonding electrons. As the absolute SE yields from our TDDFT treatment can only approximate the integrated valence ionization cross section, it is not unreasonable to expect that they should not show the same absolute magnitudes as the total ionization cross sections measured via SEEBIC. The limited goal of the simulations (beyond simply demonstrating what type of calculations may be useful for modeling the atomic-scale variations in SEEBIC contrast) is to show how capturing the ionization of bonding electrons leads to anisotropy in the rate of decay in the SEEBIC intensity with distance from the atomic centers (i.e. heightened intensity along the W-Se bonding directions). With this limited scope better delineated, we feel that the current results can be considered a qualified success.

Specifically, we have made the following clarifying changes (in bold) to the text of the abstract,

and in our discussion of the computational results in the main text:

“We find that the double Se lattice site shows higher emission than the W site, which is at odds with first-principles modelling of **valence** ionization of an isolated WSe₂ cluster.”

“Moreover, we illustrate that a model designed to **describe beam-induced ionization of valence electrons** in a simplified WSe₂ structure is insufficient for fully replicating the contrast observed from encapsulated WSe₂ leading to the conclusion that this imaging mode could possibly be used to capture subtle changes in the electron density distributions from the effects of, in this case, interlayer interactions.”

“...there are also notable discrepancies between the predicted (valence **only**) yields and the **total** SE yields observed in the experiment. For instance, the experimental intensity ratio between the W and Se sites is not reproduced by our **valence-only** simulations.”

Nevertheless, the explanations provided by the authors in their responses, and the more suggestive tone employed in the revised manuscript help clarify the position of their work. I do have additional comments:

- Figure 3(g), which is central to the discussion and the main message of this manuscript, as underlined by the authors in their responses and revisions. The histograms in Figure 3(e), which have been modified in this revised version of the manuscript, are directly linked to the intensity range of the SEEBIC maps in Figure 3(g). Considering the histograms, how is the contrast in the SEEBIC maps in Figure 3(g) affected by the current range? i.e. could the authors provide SEEBIC maps for currents in various ranges, e.g. above 0.2 pA and up to 0.8 pA. What is the meaning of having SEEBIC current below 0.0 pA (after the rescaling)?

The reason for choosing an intensity range between 0.0 and 0.2 pA should be indicated.

Providing a clear demonstration for the choice of current limits in the SEEBIC maps would help readers assess the robustness of the interpretation.

The reviewer is picking up on an important point that perhaps we were not sufficiently clear about. There are three issues intertwined here: 1) a numerical offset, 2) a small signal of interest,

and 3) a large noise signal.

The numerical offset and small signal

Because the WSe₂ layer is encapsulated in our sample, there is a background signal of unknown intensity collected along with the signal we are interested in. As an example, one could have a background of 100 pA and observe fluctuations on the order of 10 pA on top of this background. One would then want to plot these fluctuations with a scale ranging from 100 pA to 110 pA so that the scale matches the features of interest (instead of plotting from 0 to 110 pA). We have set the lower and upper bounds of the visualized range in Figure 3(g) as the minimum and maximum values contained within the data. Note, however, that the quantitative numerical value of the data points is not changed by just setting the display range. Because we know there is an offset with an unknown numerical value associated with the background signal, we opted to plot the numerical values on the color bar starting at zero with the minimum value so that the relative intensity is reported as the value above the average minimum current. In other words, we have shifted the numerical values so that they also range from minimum to maximum, and arbitrarily identified the minimum value with zero on the color bar legend. In the above hypothetical example where we have datapoints ranging from 100 to 110 pA this would be equivalent to shifting all the datapoints so that they instead range from 0 to 10 pA (setting 100 pA equal to zero). This procedure makes the displayed intensity range intuitively match the numerical values listed on the color bar and can be interpreted as intensity (or current) above the ‘background’ (or, to be exact, above the minimum).

It has come to our attention that the original word we used to describe this procedure was ‘rescaled’ and that this might imply a multiplicative scaling. We have replaced this word with ‘shifted’ as well as made additional clarifications regarding exactly what was done.

The large noise signal

The second point the reviewer brings up concerns the histograms. Because they also have absolute numerical values attached to them, the zero point should be set consistently with what is displayed in Fig. 3(g). This is what was done. However, the histograms show a representation of the raw datapoints while the images in (g) show averages of >700 samples. In the histograms, we

can see the large spread in the raw data—the width of the histograms—which is significantly larger than the difference between the means. There is much more overlap in the intensity distributions for the raw W and Se measurements than there is difference. Nevertheless, the confidence in the mean positions is much higher than the difference in the mean positions. In other words, despite the high variation in the raw data, we have obtained sufficient sampling to be very confident in the *mean* values. [Note: for a visual comparison of this difference compare the image in (b) to the images in (g).] If we were to measure only the background intensity (i.e. with the WSe₂ layer removed) we would obtain a similarly broad distribution centered approximately* at zero (which, as we describe in the figure caption, is identified with the smallest value recorded in the *averaged* tile), with half the values below zero and half above zero. The significance of some values being below zero and some being above zero is simply noise spread in the raw data.

*We use the word ‘approximately’ here because despite the care with which we have attempted to treat the data, this background intensity was not directly measured without the WSe₂ and we are using the minimum value found in the images in (g), which is a mean minimum, as our best estimate. It is possible that this is a slight overestimate and that there is still some contributed signal from the WSe₂.

We have added a few words to try to make these points clearer. This section now reads: “The intensity in the images shown in **Error! Reference source not found.**(g) has been shifted such that the minimum value in the mean image, which is our best estimate of the average background signal, is equal to zero,. This allows the image intensity to be interpreted directly as a current relative to the background. The minimum and maximum of the displayed intensity were determined by selecting the minimum and maximum values found in the mean images. In this way, the relative intensity of the images is preserved and the minimum value is set to zero. The colorbar lists the mean measured current values in pA.”

To directly answer the reviewer’s questions about the limits of the plot, here, we provide several variations and illustrate (it will be obvious) why we opted not to display the plots in this way.

For comparison purposes, this is the original plot shown in the publication.

Now, we plot the same data but with the min set to 0.2 pA and the max set to 0.8 pA as suggested by the reviewer.

Most of the data falls below this range so we see an almost perfectly uniform intensity at the minimum value. (If one is very perceptive one can pick out a few regions that just barely show something.)

We can do similarly and set the values on the low end (-0.2 to 0.0 pA).

All values are higher than this range so we see a uniform intensity at the maximum value.

We can also do less extreme examples. Here, we plot from 0.1 to 0.2 pA.

Clearly we are cutting off the lower values.

Here, we plot from 0.0 to 0.1 pA.

This cuts off the higher values.

We can also add an arbitrary numerical value to the data but still plot it in a range that extends from the minimum to the maximum values. Here, we add 10 (notice the color bar labels).

Here, we add 1000.

Notice that the *relative intensity is not changed*. The absolute *difference* between the minimum and maximum values is preserved, it is only an arbitrary offset that is changed.

This is what our scaling changes. We set the minimum value found in these images to zero so that the intensity starts at zero and increases by whatever amount *was measured* (in this case a maximum of around 0.2 pA). We plot from the minimum value (zero) to the maximum value found in the images with nothing cut off or omitted.

We conclude that the original, shown again here, is the most faithful and easily interpretable representation of the data.

We also note that the code and datasets are available on Github for full transparency of every processing step and to ensure reproducibility.

- In their additions to the revised manuscript, the authors indicate that (p. 12) “This signal is also present in EELS data, which detects energy losses to the primary beam electrons, however, many other energetic transitions are simultaneously detected with EELS. This makes the interpretation of the EELS signal more difficult.”

Could the authors clarify what they mean by “many other energetic transitions are simultaneously detected with EELS”, and “This makes the interpretation of the EELS signal more difficult.”? For the latter, individual excitations in EELS can typically be discriminated as a function of energy-loss and interpreted individually.

In EELS one detects *any* energetic transition. So, for example, interband transitions and plasmons are detected which represent the majority of the signal in low loss EELS. How much of the detected EELS signal is due to direct transfer of energy to emitted secondary electrons? This is extremely difficult to tell if one only has the EELS signal because it contains information from multiple sources. While it is certainly true that many features in the low-loss spectrum can be individually resolved and identified with particular quasiparticle excitations in the material, the electronic transitions associated with secondary electron generation are from (discrete) bound states to (a continuum of) unbound electronic states. The continuous nature of the excitations that result in ionization precludes any discrimination of specific excitations in the EELS spectrum above the ionization threshold of the material, which present as a largely-featureless, exponentially decaying loss function *modulo* the inner-shell loss edges. The authors have previously agreed with reviewers that the secondary electron yield should be closely related to the integrated EELS intensity associated with ionization events, and have revised the manuscript

to clarify this point in the previous round of reviews. However, we are compelled to emphasize here that it is ultimately not true that the totality of information on secondary electron generation is contained in the loss function. Internal reorganization of the material-bound electrons occurring after the initial beam-induced excitation (e.g., Auger processes) cannot be captured in the EELS spectrum, which reports only on the initial electronic excitations at their associated resonance energy. Thus, the EELS spectrum does not contain sufficient information to determine the absolute yield of secondary electrons. We must maintain that it is accurate to state that the SEEBIC intensity reports directly on the total yield of secondary electrons, and that the identical information would be extremely impractical (or in the limiting case of secondary electrons generated through Auger processes, impossible) to retrieve from EELS spectra. SEEBIC only detects the emitted secondary electrons and therefore does not require any signal decomposition to remove contributions from energetic transitions that do not result in ionization.. We have made this clarification in the text.

“Since STEM-SEEBIC exclusively reports on ionization events, it can provide information not usually accessible via standard EELS. The onset of ionization for the most weakly bound valence electrons in materials can overlap energetically with other non-ionizing excitations such as interband transitions and plasmonic losses, preventing discrimination of primary electron energy losses due to ionization events. Furthermore, since the primary electron energy losses report only on the initial excitations enacted through inelastic scattering, information on secondary electrons generated through internal electronic reorganization processes occurring after the initial excitation (e.g., Auger processes) is not present in the EELS spectrum. Thus, while the total secondary electron yield is provided directly via the SEEBIC intensity, the totality of information needed to produce this same quantity is not accessible with knowledge of the primary electron energy loss function alone, regardless of how it is measured/analyzed.”

Reviewer #2 (Remarks to the Author):

In this revised manuscript, the authors have addressed most concerns and significantly improved

the readability. But a key concern still exists, that is, what exactly is the origin of the atomic contrast in the SEEBIC image? In the manuscript, the authors tried to emphasize that the images do not show distinct electron orbitals, then why the concept of orbitals should be introduced? I agree with the authors that the SEEBIC image contrast comes from the electronic charge density excluding nuclear charge density. So good wording for the title may be the electron charge density or ionization cross section. Words like “Total electron orbital” or “summed electron orbitals” are weird and not well-defined. In electron microscopy especially, “orbital” is usually used when different orbitals can be distinguished such as d-orbitals from core-loss EELS. Using orbital in the work is very misleading and seems over claimed.

The origin of the atomic contrast in SEEBIC is the non-uniform emission of electrons from the sample due to the non-uniform distribution of sample electrons and their ionization probabilities. The wave functions associated with the electronic states (or, ‘orbitals’) must be invoked in certain portions of the manuscript to explain, e.g., that the total ionization cross section can be approximately understood as a sum of the orbital ionization cross sections over all of the occupied electronic states, and how this is related to the spatially-resolved ionization rates observed in our study. Moreover, we believe the term ‘electron orbitals’ evokes a clear mental image that is strongly associated with what is being measured in the manuscript, and that ultimately the distribution of the electron density is identified in quantum mechanics with the squared modulus of the electronic wave function. It is for this reason the authors had preferred the term ‘orbitals’.

In the initial reviewer response, it was clear that the reviewers had misunderstood what we were attempting to claim; a fault of our initial draft. In the subsequent revision of the manuscript, we added additional descriptors such as “total electron orbital cross section”, “summed electron orbitals”, and “electron density”. With these changes, it appears the reviewer now understands what we mean, but maintains that our revised presentation is still confusing. As the reviewer specifically objects to our continued use of the term ‘orbital’, we have removed the term ‘orbitals’ from the title in favor of ‘electron density’, as well as in certain portions of the text where this change would not harm the intelligibility of the discussion. However, it was not possible to make this substitution in every location in the manuscript, since we occasionally

needed a specific term to refer to different energetic levels and spatial distribution of the specimen electrons, and trying to do this without invoking orbitals as a basis seemed senselessly opaque. We hope some deference to the reviewer's judgment regarding our choice of terminology will be favorably received.

Reviewer #3 (Remarks to the Author):

The revised paper by Dyck et al. has addressed some of the concerns raised by the reviewers. As all reviewers objected against some form of overselling the orbital imaging aspect in this otherwise remarkable experiment, I find it difficult to accept that the current paper still has a title which claims

"Imaging of Total Electron Orbital Cross Sections".

for the following reasons:

1st: Orbitals are non observable objects and can therefore not be imaged in experiments.

The authors understand that this reviewer takes issue with our (admittedly colloquial) use of the term 'orbital' in a physical sense. In many ways we sympathize with this grievance, however we are far from the first in this field to commit this abuse. We use the term 'orbital' here in the same sense that it is used in references 9-17, where orbital imaging is repeatedly claimed and experimentally demonstrated (insofar as this is possible, given the purely mathematical nature of orbitals as a single-electron basis used to construct approximate many-electron wave functions). The main caveat we offer is that we cannot distinguish SEEBIC intensity contributed from individual different orbitals, a point which was extensively clarified in the first revision. Nevertheless, the term 'orbitals' has been removed from the title and in most of the manuscript, with the discussion now focused upon the observable quantity (electron density) and orbitals invoked only when wave function-dependent quantities are discussed. We agree with the reviewer that single electron orbitals are not observables in many-electron systems, and only represent a convenient basis for expressing the correlated many-electron wave functions. The terminology is nevertheless used ubiquitously in the relevant literature to describe electron states

in materials. In the spirit of maintaining continuity for our readership, we would strongly prefer not to be required to discontinue its use entirely in this context in the current manuscript.

2nd: we don't see the total electron scattering cross section either as it discussed now better how e.g. plasmon excitations don't contribute to the SEEBIC contrast while they would occur in EELS.

The authors are grateful to this reviewer for noting our mistake, and have removed the erroneous phrase “total electron scattering cross section” from our discussion as follows:

“In this work, we have presented the case that the atomic contrast observed in SEEBIC images can be understood as a summation of the total ionization cross sections for occupied atomically-localized electron orbitals, viewed as an image projection.”

3rd: I don't agree that the positive core of the atoms plays no role as it is this core that gives rise to the presence of all the states and their energies and determines the ionisation cross section. The fact that SE yield is sensitive to the work function further proves this point. (also the TDFT approach would not work if the cores wouldn't be present)

Indeed, the positive core of the atoms are responsible for the formation of bound electronic states in materials, and determine the equilibrium spatial distribution of the electron density. We do not feel that we have asserted anything to the contrary in noting that SEEBIC is not sensitive to the positive charge of the nucleus like DPC is, and does not depend on nuclear scattering for image formation like HAADF. The positive core of the atom is responsible for determining the local density of states that is detected with SEEBIC, but the nuclei themselves are not directly implicated in the image formation mechanism as in the case of DPC and HAADF.

A more accurate title proposal could be

"Atomic scale mapping of local ionisation cross section reveals anisotropic local density of electronic states in 2D materials"

The purpose of a title is not only accuracy but communicative reach. The suggested title is highly

accurate but may serve to restrict readership to a more theoretical domain, few proponents of which will be in a position to avail themselves of the empirical aspects of this work.

I also don't agree with the statement on line 82 that we get a direct imaging of the 'electronic structure'. In the conventional meaning of electronic structure, the band structure involves both reciprocal or real space position dependent information and the energy of bands. Here we get only the location and something related to density of states with the energy axis integrated out.

The authors agree that only certain aspects of the electronic structure are reflected in the SEEBIC signal, and appreciate the opportunity to clarify our presentation. "Electronic structure" has been changed to 'electron density' here.

While line 234 more correctly claims that "The SEEBIC images shown in Figure 3(g) represent the total ionization cross section of the material, convolved with the point spread function of the instrument."

On line 237 another attempt is made to redefine what is being measured.

"Thus, the SEEBIC image represents the total projected electron orbital cross section of the specimen--all electron orbitals summed together and projected".

It is however unclear on what we project here and what the difference is between an orbital cross section and the ionisation cross section. And also why this is suddenly related to the sum (weighted somehow?) of the electron orbitals (or their density?).

This was not an attempt to redefine what is being measured. This, and the verbiage used in other locations, is an attempt to describe our results with multiple different descriptive terminological formulations so that if one formulation is not understood by the reader, another formulation may help clarify. Previous reviewer comments suggested we were not clear, and indeed, from the previous comments it was obvious that our presentation was at fault. We have therefore added many different descriptive formulations, and the reviewer responses in this round suggest that we have largely corrected the original miscommunication, i.e. the reviewers now seem to understand clearly what we are talking about, but this reviewer now objects to the multiplicity of terminological formulations that have been added specifically to ensure clarity.

In this specific formulation, we are highlighting the fact that the STEM is a projection instrument. All images are a projection of the sample--there is no third dimension to an image like there is with the sample. Thus, a measurement at any location, here a measurement of total ionization, is a sum of the contributions from all of the occupied electronic states that overlap at that location (which extends in the z direction through the sample, a projection). This is, of course, weighted by the ionization probability of the electrons occupying each electronic state. We have attempted to make this clearer in the text. It now reads:

“Thus, the SEEBIC image represents the total electron density of the specimen contributed by each of the occupied electronic states (modulus squared) summed together, weighted by their respective ionization probabilities, and viewed as an image projection.”

It is remarkable to note that from about line 321 the paper suddenly becomes much more precise in describing these excitation processes which indicates that within the current author list there is ample experience and knowledge to significantly improve the loose statements made in the earlier paragraphs and to come up with a more precise title and descriptive text that does justice to this beautiful work, both experimentally and in terms of simulations.

We have gone back through the manuscript and attempted to again clarify any loose statements. While we want our terminology to be precise, we also want the presentation to be accessible to the general reader.

Overall, it appears that this reviewer now clearly understands what we are presenting but takes issue with some finer points of terminology which is likely carried over from our initial presentation which was *not* clear. The original misunderstanding stemmed from the use of the term ‘orbital’ insinuating that we were claiming we could distinguish SEEBIC intensity contributed by ionization of electrons occupying a specific electronic state. This is not the case and this point has been unambiguously and repetitively underscored. The reviewers’ suggested title makes clear that, despite current claims of lack of clarity and precision, the reviewer now very clearly understands what we have presented.

REVIEWERS' COMMENTS

Reviewer #1 (Remarks to the Author):

The authors clarified my points of concern and I thank them for the detailed explanations. It is possible that some information added over the course of the revisions in the main text could be better suited in the supplemental information (for instance, explanations about the interpretation of SEEBIC signal), but I leave this to the discretion of the authors. I recommend this manuscript for publication.

Reviewer #2 (Remarks to the Author):

No further comments.

Reviewer #3 (Remarks to the Author):

The authors have taken into account all comments from the referees. Although the process was lengthy, I am pleased to see the quality of the paper improved significantly.